# Schedules Optimization with the Use of Value Engineering and NPV Maximization

**Jerzy Rosłon** **, Mariola Książek-Nowak and Paweł Nowak ***

Civil Engineering Faculty, Warsaw University of Technology, 00-661 Warszawa, Poland;
j.roslon@il.pw.edu.pl (J.R.); mariola.ksiazek@il.pw.edu.pl (M.K.-N.)
* Correspondence: p.nowak@il.pw.edu.pl; Tel.: +48-22-234-6515

**Abstract:** Construction program, spatial, architectural, and structural decisions taken in the early stages of a project have a significant impact on meeting the goals and needs of the client. The use of principles and methods of value management allows for an in-depth analysis of the project assumptions from the investor's perspective and leads to the best ratio of the project's utility value/sustainability to the price of its implementation. However, analysis of literature sources allows to state that optimization of the economic value of the project takes place only in the preimplementation phase of the project. This paper presents the original concept of combining issues of construction project's utility and economic value optimization. The model enables the maximization of the utility value of the subject of the project, taking into account its economic parameters. To support the implementation of the model, a schedule optimization procedure was developed using metaheuristic algorithm. The model was demonstrated on the basis of a case study. The presented proprietary approach to optimize the construction schedule taking into account the economic and sustainability of a construction project can be used in "design and build" projects, with particular emphasis on projects managed in the sustainable Project Management system.

**Keywords:** construction project scheduling; value management; metaheuristic algorithms; sustainability; optimization

## 1. Introduction

The schedule is one of the most important planning and control tools for the construction process. It enables the presentation of the relationship between activities, their location in time, and the balancing of renewable resources (employees, equipment, and materials) and nonrenewable resources (cash) in individual construction subperiods [1]. A specific goal of construction planning may be, e.g., minimizing its duration, minimizing or ensuring the levelling of resource consumption, and maximizing the economic value of construction. In order to solve such problems, optimization methods are used, which can be divided into: accurate, heuristic, and metaheuristic [2–4]. Schedule optimization problems belong to the class of NP-hard (nondeterministic polynomial time hard), and the time needed to solve them grows exponentially with their size [5]. Therefore, the main disadvantage of using exact optimization methods is the fact that with large instances of problems, they do not allow to obtain a solution in an acceptable time [6]. In the light of the results of the literature review, metaheuristic algorithms seem to be the most promising tool for solving scheduling problems [7]. They are suitable for solving complex NP-difficult problems due to the possibility of obtaining suboptimal solutions in an acceptable time [8]. The literature describes widely the problem of project scheduling with regard to constraints in the availability of resources (resource-constrained project scheduling problem, RCPSP). A development of the RCPS problem is the problem of scheduling projects with limited availability of resources and with multimode activities (multimode resource-constrained

project scheduling problem, MRCPSP or, less frequently, MMRCPSP). The multimode of activities means the possibility of variants of the methods of performing individual activities, with each mode having an assigned duration and sustainable resource consumption requirements [9]. This approach allows for taking into account situations in which the contractor may, e.g., allocate more resources to a selected task in order to accelerate works on a building object, perform a given element in a different technology [10], or consider subcontracting individual works to various subcontractors [11]. In practice, investors (including developers) strive to minimize not only the time but also the cost of construction projects so as to maximize the rate of return on expenditure. Contractors also try to optimize these project parameters.

The relationship between time and cost is included in the problem of multimode resource-constrained time–cost trade-off (MRC-TCT RC-TCT or sometimes simply TCT) projects with limited resources and time–cost relationship) [12,13]. The issues of MRCPS, relating to construction issues, are developed in the literature on the subject. Further models are being developed to take into account the specificity of construction projects. However, the proposed solutions require constant support of IT specialists, which significantly hinders their practical application in the field of construction [13]. This class of issues does not take into account the key characteristics of a construction project for contracting authorities, such as functionality or utility of the facility. These issues are the subject of the concept of sustainable value management (SVM), related to the definition, maximization, and achievement of the best value for money projects [14].

In the current state of knowledge, the issues of value management (VM) are considered in the conceptual phase and in the planning phase of the project implementation. On the other hand, the issues of optimizing the project schedule are considered further, in the preimplementation phase. This results in the inability to maximize the value in use of the subject of a construction project, taking into account the economic parameters of the project (net present value, monthly cash flows), and the existing conditions and technological and organizational limitations of construction work. This is particularly acute in the case of "design and build" projects. Therefore, as a subject of research, a model for the optimization of the construction project schedule was developed, taking into account the concept of sustainable value management of the project subject.

Section 1 presents the subject of research. Section 2 presents the concept of value management in construction projects and lists the value creation factors recommended by renowned global organizations. Sections 3 and 4 describe methodology of research and the authors model of scheduling construction projects using the concept of value management and an extended version of the MRCPS problem with discounted cash flows, respectively. It describes the assumptions and the mathematical notation of the model. The subsequent steps of the procedure algorithm as well as the model validation and verification process are presented. Section 5 presents a case study illustrating the use of the model to optimize the construction schedule, taking into account the economic and use value of sustainable construction projects. The last section is a summary of the results obtained. It contains final conclusions and directions for further research, inspired by the findings and results of the analyzes.

## 2. Construction Projects Value Management

### 2.1. Definitions and Values

Value Management (VM) is related to defining, maximizing, and achieving the best value for money projects [14]. Their origins can be traced back to the 1940s in the value analysis (VA) method, first used by Lawrence Miles [15]. With the development of the method, it was realized that the basic control of materials, costs, and quality according to the specifications was not sufficient. It was also necessary to ensure that the products meet the investor's needs. Hence the idea to apply a more general concept of value, including costs, time, quality, knowledge, and technical competence [14]. Definition says: "Value management is a systemic, structured process of team decision making. Its goal is to achieve the highest possible value of a project or process by defining the functions necessary

to achieve value goals and implementing these functions at the lowest cost (Life Cycle Cost (LCC), according to the required level of quality and efficiency" [16].

Many papers [14,15] point out that the benefits of using VM are particularly important in the initial stage of the project. The use of the value management methodology enables an in-depth analysis of the goals and assumptions of projects from the investor's perspective. At the same time, there is a popular view that the application of the VM process in construction is often started too late to obtain maximum benefits [15].

The benefits of using a VM include, among others [14,17–21]:

- better understanding of the contracting authority's needs (including: improved communication, improved identification, assessment and risk management, better understanding of the project by stakeholders, and increased involvement of the contracting authority in works on the project);
- getting rid of unnecessary costs (including: energy saving, avoiding over-specification of the subject of the contract, improving costs in the sense of LCC, using alternative solutions and materials, and reducing unnecessary expenses);
- reduction of the project time (including: simplification of technological solutions and improvement of the probability of completing the project on time);
- improvement of communication and efficiency (including: informed decision-making, better understanding of the project, increased involvement in works, and reduction of the risk of disputes);
- improvement of the company's innovativeness (including: searching for new solutions, alternative materials, improvement of standards and company policy, and greater competitiveness on the market).

The concept of value itself seems to be widely understood, however, it can be defined differently by different construction projects stakeholders. Having no material features, it eludes measurements and attempts to quantify, despite the fact that it is one of the key and most important market concepts [14].

Numerous sources confirm that the value of the project in the sense of VM is being very hard to define, from mathematical point of view. Most often, however, it is written as [15,21,22]:

$$Value = \frac{Function}{Costs\ (LCC)} \tag{1}$$

Sometimes, a more elaborate concept of value is also introduced [17,21]:

$$Value = \frac{Function + Quality}{Costs\ (LCC)} \tag{2}$$

where:

- Function: task to be fulfilled by the object of the contract (e.g., building),
- Quality: the client's needs, expectations,
- Cost (LCC): cost in terms of the entire life cycle of the project.

*2.2. Assessment of the Facility's Ability to Perform the Required Functions*

In order to use VM techniques, it is important to develop a process for quantifying and measuring values (or functions). Such measures should be objective and unambiguous; however, it often happens that a certain dose of subjectivism cannot be completely ruled out. The general practice of the VM is to analyze each function (object/element) and determine the actual costs of its implementation and sustainable development. It is often possible to distinguish a basic function (necessary/required by the contracting authority) and additional functions influencing the value. It is very important to analyze all the functions that a given element performs in order to assess its actual value. The Chartered Institute of Building (CIOB) [14] points out that after identifying the main functions (in other words, the main

factors of creating value), it is good to establish their hierarchy, thus creating the so-called project value profile. By assigning quantitative measures (also known as metering) to each of the individual value creation factors and agreeing on metrics for function performance, the VM team can evaluate and measure performance, and thus develop a project value indicator. An example of a value profile is shown in Table 1.

**Table 1.** Sample of a value profile [14].

| Value Creation | Weight (%) | Measure | Fulfilment of a Function (1-Weak, 10-Excellent) | Weighted Result |
|---|---|---|---|---|
| Finance | 15 | Cost | 3 | 45 |
| Project management | 15 | KPI | 4 | 60 |
| Business efficiency | 20 | Score | 4 | 80 |
| Image | 10 | Research | 2 | 20 |
| Maintenance costs | 15 | Cost | 3 | 45 |
| External requirements | 10 | Audit | 8 | 80 |
| Sustainable development | 15 | Score | 6 | 90 |
| Total ratio of value | (100%) | | | 420 |

In the case of construction projects, the basic functions that the designed objects must fulfil are defined in the functional and utility programs.

### 2.3. Requirements for Construction Objects—Set of Chosen Research Criteria

The functions described are closely related to the requirements for building objects. These requirements have been grouped by leading organizations. The International Council for Building (CIB—Conseil International du Bâtiment), the International Organization for Standardization (ISO), the United Nations (UN), and also the European Economic Community (EEC) (Directive 89/10/EEC of 21 December 1988 on the approximation of laws, regulations, and administrative provisions of the Member States relating to construction products, as amended).

The structure developed by The International Council for Building (CIB) [23] is particularly interesting, namely:

1.　Safety: 1.1 Structural safety—1.1.1 Whole structure; 1.1.2 Frame/wall system; 1.1.3 Floor/diaphragm system; 1.1.4 Member; 1.1.5 Connection; 1.1.6 Foundation; 1.2 Fire safety—1.2.1 Whole building; 1.2.2 Frame/wall system; 1.2.3 Floor system/roof; 1.2.4 Other building parts (e.g., door); 1.2.5 Member/materials; 1.2.6 Services; 1.3 Safety in use—1.3.1 Whole building; 1.3.2 Frame/wall system; 1.3.3 Roof/floor system; 1.3.4 Other building parts/members/materials; 1.3.5 Services.

2.　Comfort: 2.1 Acoustical comfort—2.1.1 Whole building; 2.1.2 Frame/wall system; 2.1.3 Floor system/roof; 2.1.4 Components; 2.1.5 Connections; 2.1.6 Materials; 2.1.7 Services; 2.2 Visual comfort—2.2.1 Whole building; 2.2.2 Windows; 2.2.3 Shading devices (blinds); 2.2.4 Light caps; 2.2.5 Light shelves; 2.2.6 Wall; 2.3 Hygrothermal comfort—2.3.1 Whole building; 2.3.2 Frame/wall system; 2.3.3 Roof/floor system; 2.3.4 Member/materials; 2.3.5 Services; 2.4 Structural serviceability—2.4.1 Whole building; 2.4.2 Frame/wall system; 2.4.3 Floor system/roof; 2.4.4 Member/materials; 2.4.5 Services.

3.　Health and Hygiene: 3.1 Air quality—3.1.1 Whole building; 3.1.2 Frame/wall system; 3.1.3 Floor system/roof; 3.1.4 Components; 3.1.5 Materials; 3.1.6 Services; 3.2 Water Supply and other services; 3.3 Waste Disposal.

4.　Durability: 4.1 Structure—4.1.1 Whole building; 4.1.2 Frame/wall system; 4.1.3 Roof/floor system; 4.1.4 Member/materials; 4.1.5 Foundation; 4.2 External enclosure—4.2.1 Below ground; 4.2.2 Above ground; 4.3 Internal enclosure—4.3.1 Below ground; 4.3.2 Above ground; 4.4 Built-in furnishings and equipment; 4.5 Services.

5.  Sustainability: 5.1 Energy conservation—5.1.1 Whole building; 5.1.2 Frame/wall system; 5.1.3 Roof/floor system; 5.1.4 Foundations; 5.1.5 Components/Materials; 5.2 Green-house gas depletion—5.2.1 Whole building; 5.2.2 Structure; 5.2.3 Other parts/materials; 5.2.4 Services; 5.3 Economics; 5.4 Deconstruction/demolition and disposal.

This compilation was used by authors as criteria set, in line with the VM philosophy, focuses not on processes but on products. According to the authors, this architecture is well suited for the analysis of construction projects in the light of the management of sustainable construction projects. In particular, it can be opened for building a list of value creation factors in a table for computing a value profile (as shown in Table 1). The set of criteria used for a chosen building component is selected by experts assessing value of each task. In the construction industry, the key to success is making decisions based on the vast knowledge of experts who know the realities of the industry. Their decisions are supported by historical data and many years of experience.

As the research cited above shows, VM researches can have a very positive impact on the success of construction projects. Especially if they are used in the early stages of projects. The classic approach to value management is not, however, integrated with other aspects crucial to the success of a construction project, such as scheduling of construction works, resource constraints, and optimization of economic indicators (especially dynamic ones).

## 3. Methodology of Research

On the basis of the problem under consideration—an extended MRCPSP task in construction—the following conceptual model (the mathematical/logical/verbal representation of the problem under consideration [24]) was proposed. Its implementation on a computer is the computer model described below.

The analysis of the literature and the current situation on the construction market allowed for the validation of the conceptual model. It should be emphasized that the validation and verification of the simulation model are not unit processes and do not constitute a clearly separated stage of the simulation test. They should be treated as a continuous process taking place during the modelling cycle [25]. The scheme of the modelling process is presented in Figure 1 based on well-known research presented by R.G. Sargent (1981).

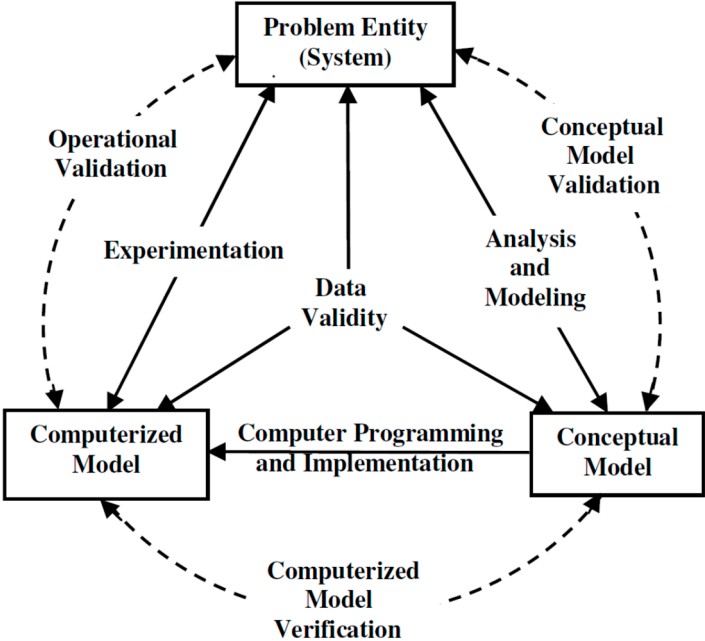

**Figure 1.** Scheme of the modelling process [25].

The verification of the computer model and the operational validation were carried out in the form of computer tests. What is important to confirm its value [26], the proposed MRCPSP model of solving complex tasks gives repeatable and satisfactory results. Additionally, part of the operational validation is presented in the case study. Data validation was carried out on the basis of the analysis of scientific literature, industry publications, and practical observations. The validation and verification of the model was performed in an iterative form [27], which means it was improved until satisfactory results were obtained.

## 4. Original Model of Scheduling Construction Projects

### 4.1. Considered Problem and Assumptions Description

The analysis of the studies and scientific works cited in the previous sections, as well as the own research conducted so far [28–33] led the authors to conviction that from the point of view of contractors in the construction industry, the net present value (NPV) indicator is the best indicator of the success of a project. The main goal of such companies is to generate profit, which should be considered over time (using dynamic methods). However, it is necessary to take into account the specificity of the market and the construction industry (including the functioning settlement systems and contracts), analyzing constraints such as construction duration (contract deadlines), technological and organizational dependencies, as well as resources (both renewable and nonrenewable and double constrained).

From the contractor's point of view [34,35], the high level of customer satisfaction has a positive effect on winning new contracts, and significantly reduces the likelihood of being involved in harmful disputes and court hearings, and incurring additional costs. If the investor decides to implement the project on his own, it is in his own interest to meet the requirements. The developed model assumes an analysis of the project and its subject in an iterative form, i.e., improving their parameters until satisfactory results are obtained or the project is finally rejected. The value analysis should start at the conceptual stage, however, in the subsequent phases of the project, it should be integrated into the schedule as quickly as possible. Usually, at the turn of the implementation planning and preimplementation phase, when the project manager is appointed, a functional and operational program and the first versions of the facility implementation schedules are created.

The warranty period is important for the construction industry. During the second stage of the closing phase of projects, it often turns out that the implementation of guarantee procedures and the loss of guarantee deposits cause a significant deterioration of the results of the implemented project, and often can even make it unprofitable. This may happen despite the fact that in the first stage of the project closure phase (immediately after completion of construction), the project showed favorable values of economic indicators and met the assumed parameters. That is why it is so important to take into account the aspects of value management, sustainable development, and maximization of performance parameters when analyzing the possibilities of maximizing the economic parameters of the project. Such procedures should start as soon as possible and proceed in an iterative form (based on feedback) as more and more detailed data about the project flow in.

### 4.2. Model

The basic mathematical record of maximizing the economic value of a construction project and the use of the subject of a construction project prepared in the process of problem modelling (assuming payments at equal intervals until the end of the project) is as follows:

$$\max F_c :$$

$$F_c = \left( \sum_{h=1}^{H+\Delta} \frac{P_h - IC_h}{(1+\alpha)^{h\,TI}} - \sum_{h=1}^{H} \sum_{m=1}^{|M_j|} \sum_{j=1}^{n} \sum_{q=\max\{t,EF_j\}}^{\min\{t+d_{jm}-1,LF_j\}} \frac{CF_{jm}}{d_{jm}\,(1+\alpha)^t}\, x_{jmq} \right) w_1 +$$

$$\left( \sum_{m=1}^{|M_j|} \sum_{j=1}^{n} \sum_{t=EF_j}^{LF_j} \frac{f_{jm} \, x_{jmt}}{J} \right) w_2 \quad, \qquad H = \left\lceil \frac{LF_j}{TI} \right\rceil, \quad t = 1, \ldots, H \tag{3}$$

$$\sum_{m=1}^{|M_j|} \sum_{t=EF_j}^{LF_j} x_{jmt} = 1, \quad j = 0, \ldots, n+1 \tag{4}$$

$$\sum_{m=1}^{|M_j|} \sum_{t=EF_i}^{LF_i} t \, x_{imt} \leq \sum_{m=1}^{|M_j|} \sum_{t=EF_j}^{LF_j} x_{jmt} \left( t - d_{jm} \right), \quad \forall (i,j) \in P \tag{5}$$

$$\sum_{j=1}^{n} \sum_{m=1}^{|M_j|} \sum_{q=\max\{t,EF_j\}}^{\min\{t+d_{jm}-1,LF_j\}} r^{\rho}_{jmk} \, x_{jmq} \leq a^{\rho}_{k}, \quad k = 1,\ldots, r^{\rho}, \, t = 1, \ldots, H \tag{6}$$

$$\sum_{j=1}^{n} \sum_{m=1}^{|M_j|} \sum_{t=EF_j}^{LF_j} r^{v}_{jml} \, x_{jmt} \leq a^{v}_{l}, \quad l = 1,\ldots, r^{v} \tag{7}$$

$$\sum_{t=EF_j}^{LF_{n+1}} t \, x_{n+1,m,t} \leq D \ , \quad j = 0, \ldots, n+1 \tag{8}$$

$$x_{jmt} \in \{0,1\}, \qquad j = 0,\ldots, n+1, \quad m \in M_j, \quad t = EF_j,\ldots, LF_j \tag{9}$$

where:

- $P_h$ and $IC_h$ are, respectively, revenues and indirect costs for the period ending on $h$, $h = 1, 2, \ldots, H$,
- $TI$ is a known time period (interval)—in the constructed $TI$ model it corresponds to one working month and is expressed in days,
- $\Delta$ is a variable that models payment delays, where payment delay is $\varepsilon$ (working days), $\Delta = \lceil \varepsilon / TI \rceil$,
- $CF_{jm}$ are cash flows related to the performance of activity $j$ in $m$ mode,
- $\alpha$ is an interest rate,
- $f_{jm}$ is the assessment of the fulfillment of the assumed functions (in meaning VM) in connection with the performance of activities $j$ in $m$ mode,
- $w_i$ are the weights of individual parts of the optimization objective function subject $\sum_{1}^{n} w_i = 1$,
- $D$ is a deadline for completion of construction.

The restriction (4) causes that each activity is performed exactly once in one of the modes. The constraint (5) is responsible for the dependencies (relations) between tasks. The limitation related to the use of renewable resources is (6) and to nonrenewable resources is (7). Doubly constrained resources can be included in (6) and (7). Constraint (8) introduces a construction completion deadline, and constraint (9) provides for consideration of binary decision variables. Due to the multicriteria nature of optimization, as well as practical aspects, including the possibility of efficient implementation in computer programs or the ability to adapt the objective function to the needs of the contractor, it was decided in this dissertation to introduce the metafunction of the objective using the aforementioned weighted sum method. This method is relatively simple and allows you to adjust the target function to specific requirements or specific priorities of the contracting company. Moreover, the multielement objective function allows for the selection of the better solution (in a general sense) in the case when two solutions are equivalent in terms of one of the maximized indicators. For example, two cases may have the same rated V value, however, one of them has a higher NPV score. Regarding the limitations, it was decided to use the aforementioned penalty functions, a solution such as that already mentioned is popular and allows for flexible shaping of the target function.

As a result of the above considerations and the research and tests carried out, the following basic objective function was proposed to be maximized during the optimization process of the initial schedules of construction projects (concept note, formula 10):

$$F_c = \underbrace{w_1 \cdot NPV_r + w_2 \cdot V_r}_{\text{objective}} \underbrace{- o_1 \cdot CF - o_2 \cdot R - o_3 \cdot T - o_4 \cdot dur}_{\text{restrictions (penalties)}} \tag{10}$$

where:

- $w_i$ are the weights of individual parts of the objective function subject to optimization—these weights highly depend on preferences of a given construction company. They are assessed and proposed by the experts based on historical data and companies' needs.
- $o_i$ are the weights of individual parts of the objective function responsible for constraints (penalties)—experts in each construction company should select proper weights according to the contractual parameters and companies' records.

The $\sum_1^n w_i = 1$, whereas $o_i$ is assigned values significantly greater than those of the first part of the objective function (goal), so that failure to meet the constraints results in disqualification of a given solution.

$NPV_r$ is the objective function component responsible for the optimization of the relative NPV value:

$$NPV_r = \frac{NPV - NPV_{min}}{NPV_{max} - NPV_{min}} \tag{11}$$

where:

- $NPV$ is the NPV value for the currently examined case,
- $NPV_{max}$ is the maximum NPV value found for the UPS (Unconstrained Project Scheduling) version of the examined example,
- $NPV_{min}$ is the minimal NPV value found for the UPS version of the examined example.

$V_r$ is a component of the objective function to include the concept of value management. It corresponds to the score obtained by a given solution in terms of meeting the functions assigned to the objects and the costs related to sustainable operation, modernization, and demolition (sustainability).

$$V_r = \frac{V - V_{min}}{V_{max} - V_{min}} \tag{12}$$

where:

- $V$ is the value rating for the currently studied case,
- $V_{max}$ is the maximum value rating found for the UPS version of the tested example,
- $V_{min}$ is the minimum value grade found for the UPS version of the tested example.

$CF$ is a binary variable responsible for meeting the condition of the maximum monthly demand for financial resources (nonrenewable).

$$CF = \begin{cases} 1 \ if \ CF > CF_{dop} \\ 0 \ in \ other \ cases \end{cases} \tag{13}$$

where:

- $CF_{dop}$ is the maximum monthly demand for financial resources allowed by the contractor.

*R* is a binary variable responsible for meeting the condition of not exceeding the maximum availability of renewable resources (i.e., working teams and construction equipment).

$$R = \begin{cases} 1 \text{ if condition } (6) \text{ is not met} \\ 0 \text{ in other cases} \end{cases} \tag{14}$$

*T* is a binary variable responsible for the fulfilment of the condition for the implementation of the entire construction in a technologically consistent manner. That means, there may be activities in the schedule for which the selection of a particular execution variant will exclude some variants in other activities.

$$T = \begin{cases} 1 \text{ if variants are not consistent} \\ 0 \text{ in other cases} \end{cases} \tag{15}$$

*dur* is a binary variable responsible for meeting the condition of not exceeding the contractual construction date.

$$dur = \begin{cases} 1 \text{ if condition } (8) \text{ is not met} \\ 0 \text{ in other cases} \end{cases} \tag{16}$$

Due to the practical aspects of computational tools in the computer model, the binary input (decision) variables were abandoned in favor of integer variables (where possible). In the proposed approach, decision variables can be divided into three types:

- Variables corresponding to variants of individual activities ($M_j$) (integer), where each variant is assigned: duration, value assessment, and demand for renewable and nonrenewable resources;
- integer variables introducing delays in the deadlines of starting tasks ($l_{ij}$);
- binary variables introducing (=1) or not (=0) additional organizational links between activities ($PO'_j$); these variables help to rank tasks.

Thus, individual problem instances formulated in this way depending on the number and range of variables may have a number of possible combinations, as in formula (17):

$$\prod_{j=1}^{n} \left[ |M_j| \cdot (l_{ij}^{max} - l_{ij}^{min} + 1) \right] \cdot 2^{\sum_{j=1}^{n} |PO'_j|} \tag{17}$$

where:

$PO'_j$ is a set of additional activity predecessors j, $po' \in PO'_j = \left\{ 0, \ldots, \left| PO'_j \right| \right\}$.

### 4.3. Procedure Algorithm/Computer Model

A block diagram of the proposed schedule optimization model is presented in Figure 2. The individual steps are performed with the support of appropriate computer programs. To begin with, an initial schedule is drawn up. The original schedule along with the functional and utility program (PFU) is used by the expert team to develop alternative solutions (variants/modes). Then, the assessment of all modes in terms of V is done. In order to create a universal and flexible value assessment system, it was decided to depart from the basic definition of value (the ratio of the function to the cost in the LCC sense) in favor of the assessment being a weighted sum of assessments of the fulfilment of individual functions and aspects related to sustainable development.

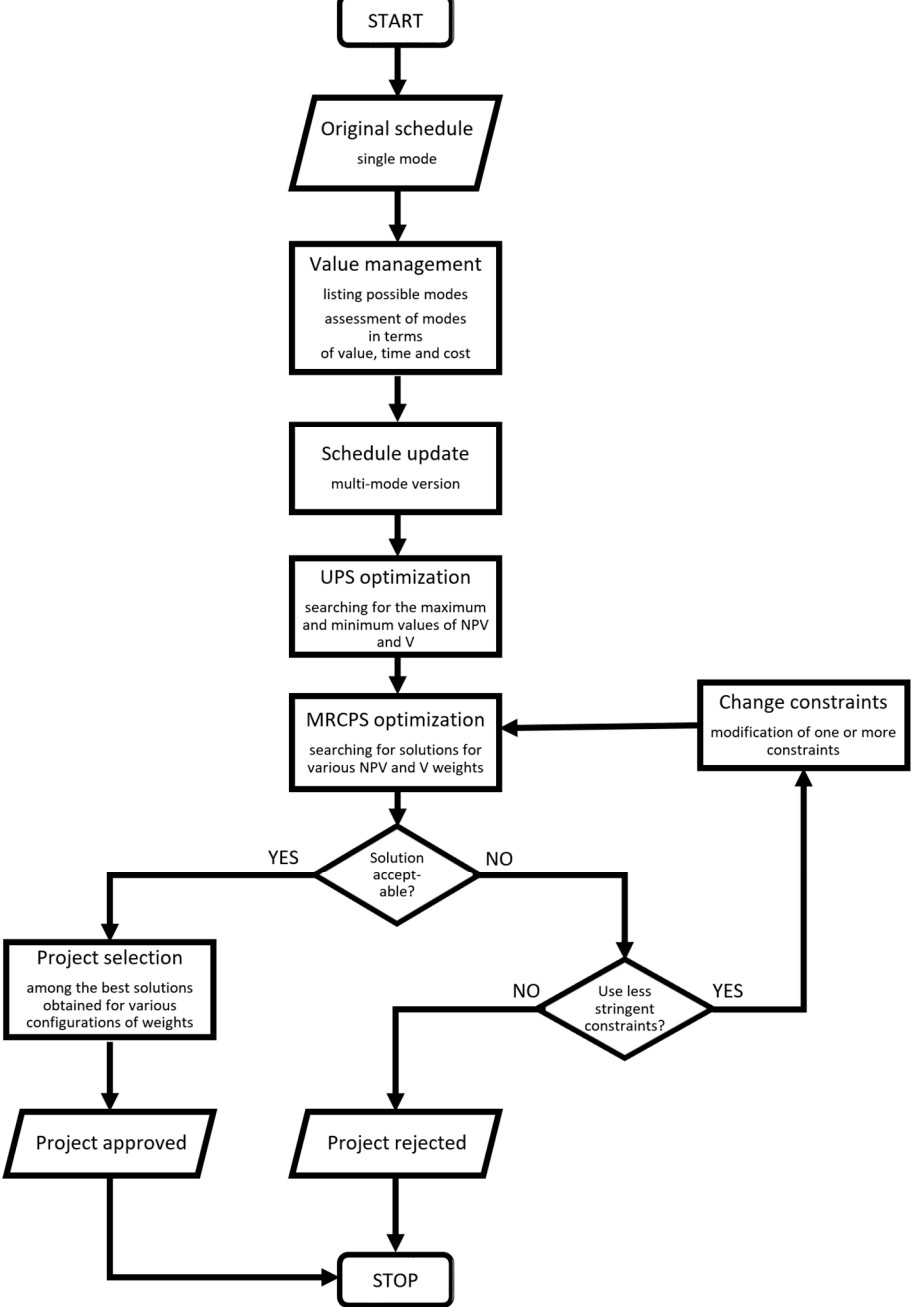

**Figure 2.** Block diagram of the proposed schedule optimization model (own source).

Thus, the assessment of individual solutions is calculated on the basis of the value profile Table using the value creation factors proposed by CIB and ISO, UN, and EEC, as it is shown in Section 2. This approach is a compromise between the flexibility of the model and the systematic nature of decisions made.

Individual factors of creating value/criteria (groups/subgroups of the procedure) got assigned weights, depending on the preferences of the decision-maker and the type of assessed activity (e.g., it does not make sense to evaluate foundation walls in terms of visual comfort, so the weight of this criterion will be defined as 0). After the tests, it was proposed to evaluate the criteria on an 11-point rating scale (from 0 to 10). However, it is also possible to use other rating systems allowing for the standardization of the final results obtained for individual variants. The assessment should be performed by a standardized team of experts working on value management procedures. The results

are normalized so that the sum of the weights is equal to 1. Thus, a vector of weights of individual factors for creating the Q value is created (formula (18), Table 2).

$$Q = [q_j] \qquad \sum_{j=1}^{n} q_j = 1 \qquad (18)$$

**Table 2.** Example of weights vector for each value-creating factor Q.

| 1 Safety | | 2 Comfort | |
|---|---|---|---|
| 1.1 Structural safety | $q_1$ | 2.1 Acoustical comfort | $q_4$ |
| 1.2 Fire safety | $q_2$ | 2.2 Visual comfort (lighting) | $q_5$ |
| 1.3 Safety in use | $q_3$ | 2.3 Hygrothermal comfort | $q_6$ |
| | | 2.4. Structural serviceability | $q_7$ |

In the next step, the same team of experts assesses individual variants in terms of all included criteria (those rated higher than 0). The solutions measurable by parameters (e.g., fire resistance class or acoustic insulation Rw (dB)) are subject to quantification. The parameters assessed by experts in a subjective manner should, in the authors opinion, be assessed using a rating scale with an odd number of grades depending on the accuracy of the possible assessment. After the tests, it was proposed to use an 11-point rating scale (from 0 to 10) or a 5-point rating scale (from 1 to 5). Thus, the evaluation matrix P is created (Table 3).

**Table 3.** Representation example of the P evaluation matrix.

| | 1 Safety | | | 2 Comfort | | | |
|---|---|---|---|---|---|---|---|
| | 1.1 Structural safety | 1.2 Fire safety | 1.3 Safety in use | 2.1 Acoustical comfort | 2.2 Visual comfort (lighting) | 2.3 Hygrothermal comfort | 2.4 Structural serviceability |
| **Variant 1** | $p_{11}$ | $p_{12}$ | $p_{13}$ | $p_{14}$ | $p_{15}$ | $p_{16}$ | $p_{17}$ |
| **Variant 2** | $p_{21}$ | $p_{22}$ | $p_{23}$ | $p_{24}$ | $p_{25}$ | $p_{26}$ | $p_{27}$ |

Variant assessments under individual criteria are then standardized. The terms of the normalized matrix $\overline{P}$ are calculated according to the formula (19).

$$\overline{p}_{ij} = \frac{p_{ij}}{\sqrt{\sum_{i=1}^{m} p_{ij}^2}} \qquad i = \overline{1, m}, \qquad j = \overline{1, n}, \qquad (19)$$

where $m$ is the number of options assessed, and $n$ is the number of value creation factors (criteria).

In the next step of the procedure, a normalized $V$ rating matrix is calculated taking into account the importance of individual criteria. The terms of the normalized matrix $V$ are calculated as follows:

$$V_{ij} = \overline{p}_{ij} \cdot q_j \, i = \overline{1, m}, \qquad j = \overline{1, n} \qquad (20)$$

The sum of the matrix components in the rows corresponding to the variants is the result of $V_i$ and is the evaluation of the individual variants in terms of value creation factors:

$$V_i = \sum_{j=1}^{n} V_{ij} \qquad i = \overline{1, m}, \qquad j = \overline{1, n} \qquad (21)$$

The results are subject to the already mentioned linear-maximum standardization, thus we obtain the V values for all modes of all activities in the schedule (the best variant has the value equal to 1). An exemplary assessment of variants was presented in the case studies (Section 5).

After completing the analysis of the values, the original schedule is updated with additional modes (including the corresponding duration, resource demand, and technological constraints T) and the calculated values of V. In the next step, for the updated schedule, the maximum and minimum NPV and V values are searched for using metaheuristic methods: $NPV_{max}$, $NPV_{min}$, $V_{max}$, and $V_{min}$. Then the schedule is updated again with the constraints on $R$ resource consumption, $CF's$ monthly cash flows, and contract duration. It is proposed to search for solutions for several different configurations of $w_i$ weights (depending on the preferences of the decision-maker). If the algorithms are unable to find feasible solutions, the constraints should be weakened and optimization should be started again. If it is not possible to further reduce the constraints, the project should be rejected. If at any point in optimization $NPV_r$ or $V_r$ reaches a value greater than 1 (very unlikely), update $NPV_{max}$, $NPV_{min}$, $V_{max}$, and $V_{min}$, and then start the optimization stage all over again.

## 5. Results—Case Study—Residential Estate Schedule Optimization

The example shows a situation in which a project manager is responsible for designing and building a housing estate. This case study analyzes and selects a variant for the construction of a housing estate consisting of three multifamily residential buildings. To begin with, an initial schedule was drawn up (Figure 3).

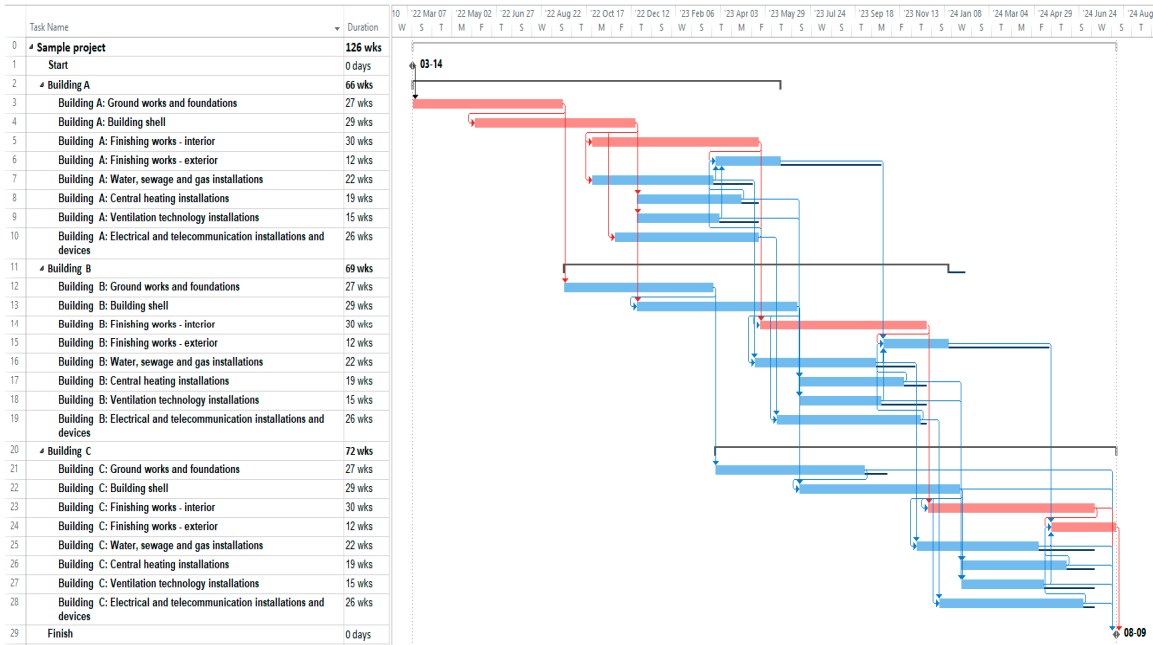

**Figure 3.** Original schedule to be optimized (own source).

Based on the original schedule and the parameters of the ordered objects, a table of variants was prepared with a description of the assessed methods (Table 4). The duration of individual activities and the dependencies between tasks were estimated on the basis of the company's historical data. In this example, the project manager limits the renewable resources (working teams and construction equipment) by organizational dependencies between the same tasks in different buildings (no possibility to carry out the same works on two objects simultaneously). However, it does not consider additional limitations on renewable resources, as it assumes hiring subcontracting companies within the financial resources (nonrenewable resources). The costs related to the individual activity implementation were estimated on the basis of data collected during the implementation of previous projects and publicly available studies. The $V_i$ values were determined on the basis of the value profile table (the evaluation of the values is presented below for the example of the item "electrical

and telecommunication installations and equipment," which also includes the installation of short transport equipment). The assessment of individual variants is presented in Table 5.

**Table 4.** Description of variants-case study (own source).

| | Task | Variant 1 | Variant 2 | Variant 3 | Variant 4 |
|---|---|---|---|---|---|
| Id | Name | Variant Description | Variant Description | Variant Description | Variant Description |
| 1 | Start | – | – | – | – |
| 2 | Building A: ground works and foundations | Heavy insulation | Waterproof concrete-"White bathtub" technology | – | – |
| 3 | Building A: building shell | Monolithic reinforced concrete frame | Reinforced concrete full monolithic structure | Prefabricated reinforced concrete structure | – |
| 4 | Building A: finishing works—interior | Standard B | Standard A | Standard B, acceleration of works | – |
| 5 | Building A: finishing works-exterior | Standard B | Standard A | Standard B, acceleration of works | – |
| 6 | Building A: water, sewage and gas installations | Standard B | Standard A | – | – |
| 7 | Building A: central heating installations | Standard B | Standard A | – | – |
| 8 | Building A: ventilation technology installations | Default standard | – | – | – |
| 9 | Building A: electrical and telecommunication installations and devices | Standard: lift A, installations A | Standard: lift A, installations B | Standard: lift B, installations A | Standard: lift B, installations B |
| 10–17 | Building B: same tasks as in building A | Same variants for building B as in building A | | | |
| 18–25 | Building C: same tasks as in building A | Same variants for building C as in building A | | | |
| 26 | End of works | – | – | – | – |

The assessment of the value of activity 9 (and at the same time 17 and 25) assumed the possibility of making electrical and teletechnical installations, as well as handling equipment in 2 quality variants-standards (a total of 4 combinations): A standard elevator and A standard installations (variant 1–V1); elevator A, installations B (variant 2–V2); elevator B, installations A (variant 3–V3); and elevator B, installations B (variant 4–V4). According to the degree of aggregation of works in the schedule, it was decided to use the value creation factors proposed by ISO, UN, and EEC. The value profile table along with the significance of individual criteria are presented below (Table 6).

According to the opinion of the expert team and after the normalization, a vector of weights was obtained for the individual factors creating the Q value (Table 7).

**Table 5.** Score of variants-case study (own source).

| Task | | Variant 1 | | | Variant 2 | | | Variant 3 | | | Variant 4 | | |
|---|---|---|---|---|---|---|---|---|---|---|---|---|---|
| Id | Id | Cost [1000 EUR] | Time t | Value V | Cost [1000 EUR] | Time t | Value V | Cost [1000 EUR] | Time t | Value V | Cost [1000 EUR] | Time t | Value V |
| 1 | Start | 0 | 0 | 1.00 | – | – | – | – | – | – | – | – | – |
| 2 | Building A: ground works and foundations | 1895 | 135 | 1.00 | 1668 | 120 | 0.90 | – | – | – | – | – | – |
| 3 | Building A: building shell | 6738 | 145 | 1.00 | 5980 | 135 | 0.97 | 7774 | 100 | 0.93 | – | – | – |
| 4 | Building A: finishing works—interior | 1946 | 150 | 0.90 | 2102 | 170 | 1,00 | 2003 | 145 | 0.90 | – | – | – |
| 5 | Building A: finishing works —exterior | 1518 | 60 | 0.93 | 1702 | 70 | 1,00 | 1612 | 55 | 0.92 | – | – | – |
| 6 | Building A: water, sewage, and gas installations | 752 | 110 | 0.89 | 1163 | 125 | 1.00 | – | – | – | – | – | – |
| 7 | Building A: central heating installations | 652 | 95 | 0.96 | 678 | 100 | 1.00 | – | – | – | – | – | – |
| 8 | Building A: ventilation technology installations | 106 | 75 | 1.00 | – | – | – | – | – | – | – | – | – |
| 9 | Building A: electrical and telecommunication installations and devices | 2233 | 130 | 1.00 | 1363 | 105 | 0.82 | 1921 | 125 | 0.85 | 1051 | 100 | 0.67 |
| 10– 17 | Building B: same tasks as in building A | | | | Same values for building B as in building A | | | | | | | | |
| 18– 25 | Building C: same tasks as in building A | | | | Same values for building C as in building A | | | | | | | | |
| 26 | End of works | 0 | 0 | 1.00 | – | – | – | – | – | – | – | – | – |

**Table 6.** Value profile table (own source).

| | 1 Safety | | | 2 Comfort | | | | 3 Health and Hygiene | | | 4 Stability | 5 Sustainable Development | | | |
|---|---|---|---|---|---|---|---|---|---|---|---|---|---|---|---|
| | 1.1 Safety of structure | 1.2 Fire safety (fire resistance (min)) | 1.3 Safety of use | 2.1 Acoustic comfort (acoustic insulation Rw (dB) | 2.2 Visual comfort (lighting) | 2.3 Hygrothermal comfort | 2.4 Utility | 3.1 Air quality | 3.2 Water supply and other utilities | 3.3 Waste disposal | 4.1 Sustainable | 5.1 Energy saving | 5.2 Greenhouse gas emissions | 5.3 Economics (running costs) | 5.4 Dismantling and utilization |
| Rating criterion | 0 | 10 | 10 | 2 | 6 | 0 | 3 | 0 | 0 | 2 | 10 | 8 | 0 | 10 | 4 |
| V1 | 1 | 10 | 10 | 10 | 10 | 1 | 10 | 1 | 1 | 7 | 10 | 7 | 1 | 10 | 9 |
| V2 | 1 | 7 | 8 | 10 | 7 | 1 | 10 | 1 | 1 | 10 | 9 | 5 | 1 | 9 | 6 |
| V3 | 1 | 9 | 7 | 8 | 10 | 1 | 8 | 1 | 1 | 6 | 7 | 10 | 1 | 6 | 10 |
| V4 | 1 | 6 | 5 | 8 | 7 | 1 | 8 | 1 | 1 | 9 | 6 | 8 | 1 | 5 | 7 |

**Table 7.** Illustrative representation of the vector of weights for each factor creating Q value (own source).

| Criterion | Weight |
|---|---|
| 1.1 Safety of structure | 0 |
| 1.2 Fire safety | 0.153846 |
| 1.3 Safety of use | 0.153846 |
| 2.1 Acoustic comfort | 0.030769 |
| 2.2 Visual comfort (lighting) | 0.092308 |
| 2.3 Hygrothermal comfort | 0 |
| 2.4 Utility | 0.046154 |
| 3.1 Air quality | 0 |
| 3.2 Water and other utilities supply | 0 |
| 3.3 Waste disposal | 0.030769 |
| 4.1 Durability | 0.153846 |
| 5.1 Energy saving | 0.123077 |
| 5.2 Emission of greenhouse gases | 0 |
| 5.3 Economics (operating costs) | 0.153846 |
| 5.4 Dismantling and utilization | 0.061538 |

After normalization, we obtain a normalized evaluation matrix (Table 8).

**Table 8.** Normalized assessment matrix $\overline{P}$ (own source).

| | 1.1 | 1.2 | 1.3 | 2.1 | 2.2 | 2.3 | 2.4 | 3.1 | 3.2 | 3.3 | 4.1 | 5.1 | 5.2 | 5.3 | 5.4 |
|---|---|---|---|---|---|---|---|---|---|---|---|---|---|---|---|
| W1 | 0.500 | 0.613 | 0.648 | 0.552 | 0.579 | 0.500 | 0.552 | 0.500 | 0.500 | 0.429 | 0.613 | 0.454 | 0.500 | 0.643 | 0.552 |
| W2 | 0.500 | 0.429 | 0.519 | 0.552 | 0.405 | 0.500 | 0.552 | 0.500 | 0.500 | 0.613 | 0.552 | 0.324 | 0.500 | 0.579 | 0.368 |
| W3 | 0.500 | 0.552 | 0.454 | 0.442 | 0.579 | 0.500 | 0.442 | 0.500 | 0.500 | 0.368 | 0.429 | 0.648 | 0.500 | 0.386 | 0.613 |
| W4 | 0.500 | 0.368 | 0.324 | 0.442 | 0.405 | 0.500 | 0.442 | 0.500 | 0.500 | 0.552 | 0.368 | 0.519 | 0.500 | 0.321 | 0.429 |

In the next step of the procedure, a normalized V rating matrix was calculated, taking into account the importance of individual value creation factors (Table 9).

**Table 9.** Assessment matrix V (own source).

| | 1.1 | 1.2 | 1.3 | 2.1 | 2.2 | 2.3 | 2.4 | 3.1 | 3.2 | 3.3 | 4.1 | 5.1 | 5.2 | 5.3 | 5.4 |
|---|---|---|---|---|---|---|---|---|---|---|---|---|---|---|---|
| W1 | 0.000 | 6.131 | 6.482 | 1.104 | 3.476 | 0.000 | 1.656 | 0.000 | 0.000 | 0.858 | 6.131 | 3.630 | 0.000 | 6.428 | 2.207 |
| W2 | 0.000 | 4.292 | 5.186 | 1.104 | 2.433 | 0.000 | 1.656 | 0.000 | 0.000 | 1.226 | 5.518 | 2.593 | 0.000 | 5.785 | 1.472 |
| W3 | 0.000 | 5.518 | 4.537 | 0.883 | 3.476 | 0.000 | 1.325 | 0.000 | 0.000 | 0.736 | 4.292 | 5.186 | 0.000 | 3.857 | 2.453 |
| W4 | 0.000 | 3.679 | 3.241 | 0.883 | 2.433 | 0.000 | 1.325 | 0.000 | 0.000 | 1.104 | 3.679 | 4.149 | 0.000 | 3.214 | 1.717 |

As a result of summation and standardization, the final scores for the individual variants were obtained. They are shown in Table 10.

**Table 10.** Assessment of V variants for activity 9 (own source).

| Variant | Score V |
|---|---|
| W1 | 1.000 |
| W2 | 0.821 |
| W3 | 0.847 |
| W4 | 0.667 |

### 5.1. Schedule Update

After performing a value analysis, the original schedule was updated with additional information, including durations and resource requirements. Additionally, the calculated values of V were introduced and data concerning the contractual period, 130 weeks, were introduced. The penalty for exceeding the deadline was EUR 20,000 for a week of delay. Indirect costs are also included.

### 5.2. UPS Optimization

In the next step, for the updated schedule, a metaheuristic algorithm was used (the case study used a commercial algorithm based on search forbidden movements included in the OptQuest® Engine, OptTek Systems, Inc.'s package) for the maximum and minimum values of *NPV* and *V*: $NPV_{max}$, $NPV_{min}$, $V_{max}$, and $V_{min}$. The results are presented in Table 11. The analyzed variables were variants of works execution (from 1 to 4 possible options) and delays for individual work packages (from 9 to 58 possible options depending on the activity).

**Table 11.** Determined extreme values of net present value (NPV) and V (UPS optimization) (own source).

| Indicator | Value |
|:---:|:---:|
| $NPV_{max}$ | 2 794 730 EUR |
| $NPV_{min}$ | 1 419 593 EUR |
| $V_{max}$ | 1.000 |
| $V_{min}$ | 0.896 |

### 5.3. MRCPS Optimization and Design Selection

Optimization of MRCPS laboratories for fifty weights: $w_1 = 0.8$ and $w_2 = 0.2$, $w_1 = 0.65$ and $w_2 = 0.35$, $w_1 = 0.5$ and $w_2 = 0.5$, $w_1 = 0.35$ and $w_2 = 0.65$, and $w_1 = 0.2$ and $w_2 = 0.8$.

The optimization results for different sets of weights are shown in Figure 4. These results constitute a set of alternative versions of the project. The decision-maker should choose one or reject the project. As mentioned earlier, one of the multicriteria decision-making methods can be used in the final selection. In this case, the decision-maker decided to choose the variant with the highest V value (the one that corresponds to the highest correlated NPV value—NPV = 0.35 and V = 0.65 weights). An illustrative schedule of the selected variant is presented in Figure 5. As one can see, comparing Figures 3 and 5, the critical path was changed. In addition, the duration of the selected schedule is longer. However, minimization of time was not the optimization criterion. The goal of the optimization was to maximize NPV and value V of the construction project. This goal was achieved.

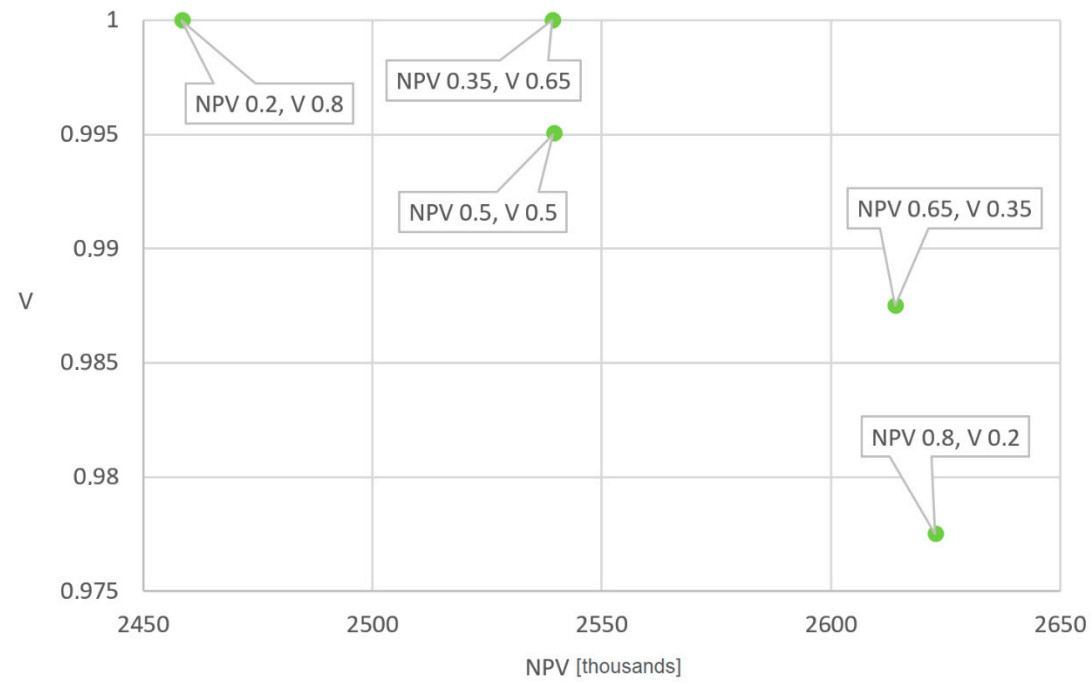

**Figure 4.** Optimization results [own calculations].

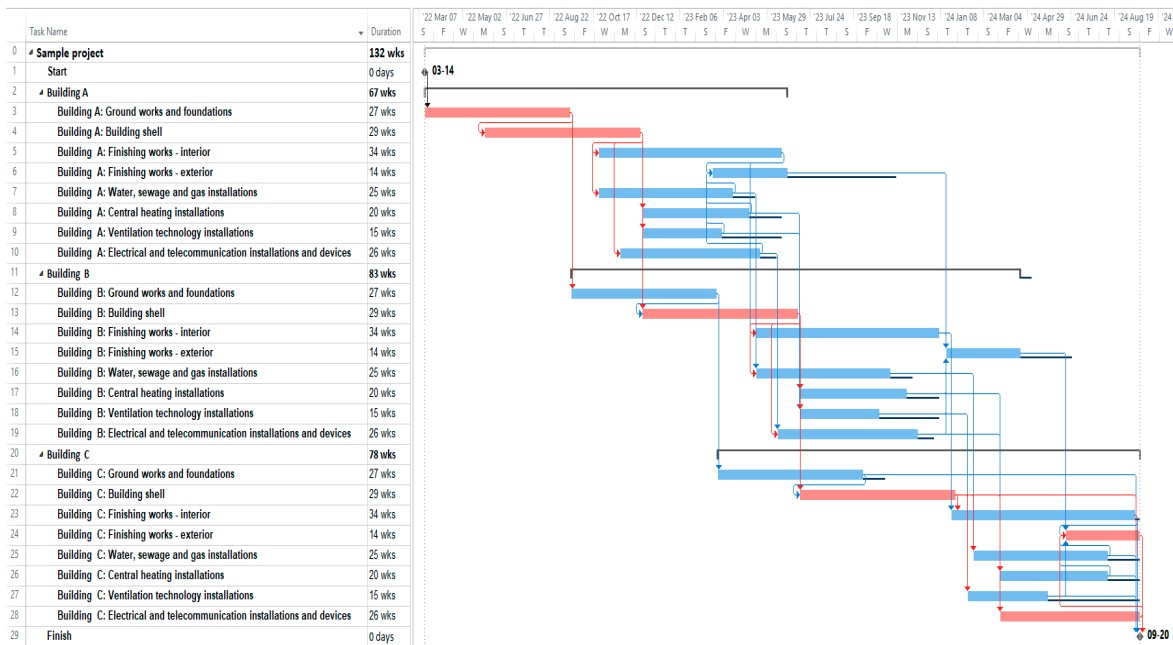

**Figure 5.** Selected schedule after optimization (own source).

As a result of the optimization procedure, NPV and V parameters were significantly improved. Original parameters: NPV = 2,183,906 and V = 0.960. Optimized parameters: NPV = 2,539,120 and V = 1.000. As shown by the obtained results, the method in use has a high utility potential.

## 6. Conclusions

So far in construction practice, value management and NPV maximization for schedules were treated separately. This approach is particularly unfavorable in the case of projects which are designed and implemented by one economic entity. The presented proprietary approach to the optimization of the construction schedule, taking into account the economic value of the project and

the useable value of the subject of the construction project, can be used in projects of the "design and build" type, with particular emphasis on projects managed in the Project Management—PM system (contract management). The proposed model can also be used in the Construction Management—CM (contracting management) system, in which the consulting company managing the investment tries to maximize the value of the project in accordance with the investor's expectations, while taking into account parameters such as: NPV, monthly flows, and time constraints. The model can also be applied to the combined PM/CM system.

The proposed approach involves the use of value management elements for better evaluation of construction projects. As mentioned earlier, the classic approach to MRCPSP (even in the extension to MRCPSPDCF) does not sufficiently take into account the key aspects for construction and the investor's needs. On the other hand, the VM methodology does not refer to the time, construction conditions, and most of all to dynamic economic indicators, key to the company's financial success.

**Author Contributions:** Conceptualization, J.R., M.K.-N. and P.N.; formal analysis, J.R.; methodology, J.R., M.K.-N. and P.N.; project administration, P.N.; writing—original draft, J.R., M.K.-N. and P.N.; writing—review and editing, J.R., M.K.-N. and P.N.; supervision, P.N.; All authors have read and agreed to the published version of the manuscript.

**Funding:** This research received no external funding.

**Conflicts of Interest:** The authors declare no conflict of interest.

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
