# Peer review of "Schedules Optimization with the Use of Value Engineering and NPV Maximization"

_sustainability, doi:10.3390/su12187454_

Round 1

Reviewer 1 Report

Figure 5. can not show the improvements from the original schedule Figure 3. The authors should demonstrate the differences between them.

How to keep the weights objective for better optimization results? The authors should provide more discussions about it.

It is a multi-criteria problem. How are the criteria identified? BY experts or historical data?

Author Response

Many thanks – all reviewer 's  comments  were taken under consideration. English improved by native speaker. Some minor editing was done. Figure 3 and 5 – differentiated and described in more proper way. Discussion about weights objectives was developed. Criteria identification method was described in the more clear way.

ALL CHANGES ADVISED BY REVIEWER 1 – marked in RED in the newly uploaded article.

Reviewer 2 Report

The paper deals with the theme of the value management in the field of buildings construction, trying to develop a model which take into account construction project’s utility and economic value optimization.

Starting from a traditional approach to evaluating time and cost of the construction, a model for the optimization of the construction project schedule was developed, taking into account the concept of sustainable value management of the project subject.

In order to develop a process for quantifying and measuring values (or functions) of the buildings, the requirements for construction objects has been introduced in the model. They include: safety, comfort, health and hygiene, durability and sustainability.

The developed model is interesting, but discussion and conclusions need to be improved to clarify the utility of the purposed approach. Particularly, the idea of implementing a quantitative approach with a qualitative evaluation need to have a further development.

I also suggest to change the title of the paper for a better comprehension of the content.

Minor revision is required

Author Response

Many thanks – all reviewer 2 comments  were taken under consideration. English improved by native speaker. Some minor revisions were done. Title slightly changed from:

Schedules Optimization with the use of Value Engineering

to

Schedules Optimization with the use of Value Engineering and NPV maximization

ALL CHANGES ADVISED BY REVIEWER 2 – marked in BLUE in the newly uploaded article.

Reviewer 3 Report

After reading the manuscript “Schedules Optimization with the use of Value Engineering”,  I highlight next major comments:

  • The Abstract reflects two statements in lines 11-16 over which the reviewer strongly disagrees. The key objective of value engineering is the optimization of schedule and budget by preserving technical project requirements, sustainability is not targeted. Proposals involving value engineering approaches are also suggested in the execution phase upon the request of contractor. The term “pre-implementation phase of the project” is ambiguous. Furthermore, a defined research aim was not provided, and main results of the study were not summarized in this section either.
  • The introduction section aims at “selling” the study to readers, that is why a background (“for what?”) that underpin the need of developing the article, a defined research aim, a brief description of results and key contributions in the field should be highlighted. Instead, authors presented a set of unrelated arguments that might be used in the literature review. For instance, lines 36 and 37 should be widely developed and correlated with the previous text.
  • Abundant transliterated information was provided in the second section with scarce interest and value for readers due to its descriptive nature. Literature review should be focused on contextualizing the topic addressed by the study, through the characterization and correlation of main concepts and the identification of gaps to be bridged. At this point, a clear definition of the research goal is crucial to progress in the right way.
  • It is very debatable the linkage suggested by authors between value engineering and the initial project phase. The preparation of the project demands the enforcement of requirements and constraints established by the client, mainly in terms of technical specifications, budget and schedule. Hence, the mission of technical teams is the fulfillment of them. The application of the notion “value engineering” to that is in fact controversial. However, this process makes sense during the execution time.
  • Since the objective of the study was not determined, the purpose of the given method presented in Figure 1 with no further explanation is also unclear.
  • Section 4.1 is irrelevant because of its weak connection with value engineering subject. Definition of all variables used in section 4 should be disclosed, i.e. what is Fc in line 251? Why are there two functions Fc (lines 251 and 291)? Reasons why Figure 1 and Figure 2 reveal two different models of project optimization are unknown, maybe an accurate combination of sections 3 and 4 is more suitable to depict the methodology followed, since most section 4 involves a given mathematical model, whose rationale and bases are unknown.
  • The description of the algorithm posed in lines 360-370 is very vague and hardly replicable. Four generic frameworks were considered for value creation factors without specifying which of them is the most suitable and why. The assessment of weightings from experts scores was not revealed either. Moreover, the use of expert assessment to determine weightings casts into doubt the whole research because experts judgment could serve to directly appraise all alternatives to select the most suitable without any algorithm. The process to transform outputs of the updated schedule to variables to iterate later (lines 400-402) is unclear, as well as criteria to stop the optimization process.
  • No conclusion was summarized as result of research findings. Limitations and contributions encountered during the development of the study were not highlighted either.
  • Miscellaneous comments. Some typewriting mistakes were found. English grammar and style should be significantly enhanced. Abbreviations must be defined at first appearance (NPV).

The manuscript presents serious flaws due to the lack of a defined goal and an unclear methodology that impede the replicability. Furthermore, the methodology proposed is highly controversial with diverse gaps. However, the concept and application of “value engineering” should be reviewed to be properly contextualized because bases on which the study is grounded are very debatable. Indeed, the methodology proposed is very distant from the reality of projects. As stated above, the role played by experts casts into doubt the value of the study, since they can determine without any algorithm which alternative is the most suitable in terms of budget and time, by preserving required technical specifications.

Author Response

We agree with the academic reviewer that the suggestions of reviewers 1 and 2 were particularly valuable and accurate. At the same time, we completely and strongly disagree with the main statement from reviewer 3 – that sustainability has nothing in common with value engineering, as it relates to time and costs only. In modern world of construction projects, changing the technical aspects of the project during „execution time” is the worst idea, (but as such it is strongly recommended by reviewer 3). Such a changes of course happens very often, but usually cause disputes, and results in rise of costs and extension of project time.  

Additionally we addressed few reasonable comments from reviewer 3:

- clear definition of the research aims were underlined

- figure 1 explained

- section 4.1. formulas connection with VE was given,  mathematical model rationale explained shortly for an VE/VM beginners

- why experts opinions should be considered as part of the algorithm was explained as well

- extremely well known abbreviations, like NPV, were explained for the reviewer 3.

ALL CHANGES DONE, as ADVISED BY REVIEWER 3 – marked in yellow in the newly uploaded article.